# Effect of Recombinant Human Erythroferrone Protein on Hepcidin Gene (*Hamp1*) Expression in HepG2 and HuH7 Cells

**DOI:** 10.3390/ma14216480

**Published:** 2021-10-28

**Authors:** Min Min Than, Pimpisid Koonyosying, Jetsada Ruangsuriya, Sunhawit Junrungsee, Chairat Uthaipibull, Somdet Srichairatanakool

**Affiliations:** 1Department of Biochemistry, Faculty of Medicine, Chiang Mai University, Chiang Mai 50200, Thailand; mmthan83@gmail.com (M.M.T.); pimpisid_m@hotmail.com (P.K.); jetsjets@hotmail.com (J.R.); 2Department of Biochemistry, University of Medicine, Mandalay 05021, Myanmar; 3Department of Surgery, Faculty of Medicine, Chiang Mai University, Chiang Mai 50200, Thailand; sunhawit.j@cmu.ac.th; 4Protein-Ligand Engineering and Molecular Biology Laboratory, National Center for Genetic Engineering and Biotechnology, National Science and Technology Development Agency, Thailand Science Park, Pathum Thani 12120, Thailand; chairat.u@tcels.or.th

**Keywords:** erythroferrone, hepcidin, iron, recombinant protein, receptor

## Abstract

Iron is essential for all living organisms. It is strictly controlled by iron transporters, transferrin receptors, ferroportin and hepcidin. Erythroferrone (ERFE) is an iron-regulatory hormone which is highly expressed in erythroblasts by erythropoietin (EPO) stimulation and osteoblasts independently of EPO by sequestering bone morphogenetic proteins and inhibiting hepatic hepcidin expression. Although the hepcidin suppressive function of ERFE is known, its receptors still require investigation. Here, we aim to identify ERFE receptors on the HepG2 and Huh7 cells responsible for ERFE. Recombinant ERFE (rERFE) was first produced in HEK293 cells transfected with pcDNA3.1 + ERFE, then purified and detected by Western blot. The liver cells were treated with an rERFE-rich medium of transfected HEK293 cells and a purified rERFE-supplemented medium at various time points, and hepcidin gene (*Hamp1*) expression was determined using qRT-PCR. The results show that 37-kD rERFE was expressed in HEK293 cells. *Hamp1* was suppressed at 3 h and 6 h in Huh7 cells after rERFE treatments (*p* < 0.05), then restored to the original levels. *Hamp1* was activated after treatment with purified rERFE for 24 h and 48 h. Together, these results reveal that ERFE suppressed *Hamp1* expression in liver cells, possibly acting on membrane ERFE receptor, which in Huh7 cells was more sensitive to the ERFE concentrate.

## 1. Introduction

Erythroferrone (ERFE) is encoded by the *ERFE* gene located on chromosome 2q37.3 (9918 nucleotides long), and was first identified as myonectin or complement C1q tumor necrosis factor related protein 15 (CTRP15) [1,2]. It is an erythroid iron-regulatory hormone which is produced in response to erythropoietin (EPO) stimulation in the bone marrow and the spleen via JAK2- STAT5 [3,4]. ERFE mainly has a hepcidin suppressive effect on hepatocytes, as well as a novel myokine which promotes lipid uptake into adipocytes and hepatocytes via transcriptional upregulation of genes involved in fatty acid uptake [5,6]. In β-thalassemia intermedia mice, ERFE mRNA expression is increased 10-fold in the bone marrow and 16-fold in the spleen; it suppresses hepcidin production and causes pathological iron absorption and iron overload [1,7].

Human ERFE contains 354 amino acids, with *N*-glycosylated Asn-229 and Asn-281 residues, Cys-273, Cys-278, Cys-142 and Cys-194 residues forming disulfide linkages or heterometric complexes with C1q/tumor necrotic factor (C1qTNF) and family with sequence similarity 132, member A (FAM132A)-related proteins [1,8]. Though it has four consensus sequences (N-X-Ser/Thr) for potential *N*-linked glycosylation sites, three glycosylation sites were found at amino acid positions 243, 295 and 333 [8]. Importantly, expression of hepatic hepcidin can be influenced by ERFE, which may suppress hepcidin expression during instances of stress-related erythropoiesis (such as acute blood loss, iron-deficiency anemia and anemia associated with chronic diseases). Moreover, it may also function to inversely upregulate hepcidin expression during iron overload. Previous studies have shown that ERFE did not use bone morphogenic protein (BMP)/SMAD pathway to suppress hepcidin gene transcription, but rather utilized the new downstream effector signaling molecules by binding to ERFE receptors [9,10]. In contrast, Arezes and colleagues have demonstrated that EPO suppressed hepcidin and other BMP target genes in a dose-dependent manner, as well as the hepatic BMP subgroup (e.g., BMP5, BMP6 and BMP7)/SMAD pathway, independently of plasma and hepatic iron levels [11]. In addition, they have confirmed that BMPs as well as anti-ERFE antibodies bind to ERFE receptors, consequently preventing hepcidin suppression and decreasing iron content in ERFE-treated Huh7 cells [12]. Jaratsittisin and colleagues found early activation followed by later suppression of the hepcidin gene (*Hamp1)* in HepG2 cells after treatment with a conditioned medium of differentiating human erythroblast culture [13]. However, both the receptor and the signaling pathway of ERFE still need to be identified.

Although primary liver cells are the best means for identification of this receptor, and the limitations on new passaging imposed by ethical consideration represent a major hindrance. Thus, liver cell lines such as human hepatocellular carcinoma (HepG2 and Huh7) cells that respond to ERFE treatment should be well-screened to identify this receptor as well as cell signaling. In this study, we aimed to produce the recombinant erythroferrone (rERFE) protein in the HEK293 cell line and investigate which liver secondary cell line is more suitable for rERFE-treated experiments, either via its receptor or through cognate protein identification.

## 2. Materials and Methods

### 2.1. Chemicals and Reagents

Human embryonic kidney 293 (HEK293) and human hepatocellular carcinoma (HepG2 and HuH7) cell lines were purchased from the American Type Culture Collection (ATCC), Manassas, VA, USA. Dulbecco’s Modified Eagle Medium (DMEM), serum-reduced Minimal Essential Medium (Opti-MEM™), Tris-acetate-ethylene diamine tetraacetic acid (TAE) pH 8.0 buffer containing 40 mM Tris, 20 mM acetic acid, and 1 mM ethylenediamine tetraacetic acid (EDTA), trypsin-EDTA solution, Hank’s balanced salt solution (HBS), phosphate-buffered saline pH 7.0 (PBS), Ponceau-S dye solution, rabbit anti-DYKDDDDK (FLAG) IgG antibody and goat anti-rabbit IgG antibody conjugated with horse-radish peroxidase (HRP) were purchased from Sigma-Aldrich Chemicals Company, Saint’s Louis, MO, USA. Pierce™ enhanced chemiluminescence (ECL) Western blotting substrate was purchased from ThermoFisher Scientific Corporation, Waltham, MA, USA. l-Ascorbic acid, bovine serum albumin (BSA), fetal bovine serum (FBS), calcium chloride (CaCl_2_), dimethysulfoxide (DMSO) and 10% sodium dodecylsulfate-polyacrylamide (SDS-PAGE) slab gel were purchased from Sigma-Aldrich Chemicals Company, Saint’s Louis, MO, USA. Plasmid pcDNA3.1 + that contains human erythroferrone gene (ERFE, FAM132B) with FLAG tag (pcDNA3.1 + ERFE) was purchased from OriGene Biotechnology Company, Rockville, MD, USA. Nitrocellulose membrane and Bradford’s solution were purchased from Bio-Rad Laboratories, Inc., Hercules, CA, USA.

### 2.2. Cell Culture

HEK293 cells were seeded in a cell culture dish (90 mm × 20 mm) in 10–15 mL of complete DMEM to <50% confluency of cell density on the day of transfection. At 3–4 h before transfection, the old culture media were replaced with new complete DMEM [14]. HepG2 cells and Huh7 cells were first seeded in 12-well plate with complete DMEM. On the following day, the old complete DMEM were replaced with incomplete DMEM supplemented with 1% each of penicillin/streptomycin and l-glutamine for 24 h [13,15]. The cells were maintained by a subculturing process when 80–90% confluence was reached. The medium supernatant was discarded and the cells were washed with pre-warmed 1× PBS. Then, the cells were trypsinized with 0.25% trypsin-EDTA solution (1 mL for T-25 flask and 2 mL for T-75 flask) at 37 °C for 5 min. The trypsinization was stopped by adding an equal volume of pre-warmed culture medium. The cell suspension was centrifuged at 200× *g* for 5 min. After the supernatant was discarded, the pellet was re-suspended in 5 mL of pre-warmed culture medium. The cell solution was diluted with an appropriate amount of pre-warmed culture medium to the required number of the cells for the next experiments by counting with a hemocytometer. Finally, those cells were transferred into a new flask containing either pre-warmed culture media (8 mL for T-25 flask and 15 mL for T-75 flask) or the experiment culture dishes. The cells were incubated in CO_2_ incubator at 37 °C with 5% CO_2_ atmospheric condition.

The number of the cells was determined using a hemocytometer and counted under the microscope. To determine the viable cells, the cell suspension was diluted in 1× PBS and mixed with 0.4% trypan blue solution (1:2 dilution). A drop of cell suspension was added to the edge between the coverslip and a slide. The number of cells was counted in several of the 1 mm × 1 mm squares and calculated as cells/mL unit.
Cells per mL = Average number of the cells × dilution factor (2) × 10^4^

### 2.3. Expression of Recombinant Erythroferrone (ERFE–FLAG) Protein in HEK293 Cell Line

Recombinant erythroferrone-tagged Flag^®^ (DYKDDDDK) (rERFE-FLAG) protein was expressed in HEK293 cells using the method previously established by Tan and colleagues [16]. First, plasmid DNA (pcDNA3.1 + ERFE-FLAG or pcDNA3.1+) (10 μg) was diluted in 1× TAE pH 8.0 to make a final volume of 450 μL. Then, 2.5 M CaCl_2_ pH 7.2 solution (50 μL) was added into diluted DNA plasmid. Those DNA-calcium mixtures were added dropwise by pipette into 2× HBS while 2× bubbling HBS. The dish was gently rocked back-and-forth and side-by-side and incubated at 37 °C in a 5% CO_2_ incubator for 24 h. Then, HEK293 cells (2 × 10^6^ cells) were washed twice with 1× PBS and were shocked with 10% DMSO for 2.5 min for greater transfection efficiency, and incubated again in complete DMEM for 48 h. Afterward, the medium was replaced with serum-free Opti-MEM medium supplemented with l-ascorbic acid (0.1 mg/mL). Finally, the culture medium was collected and pooled every 48 h three times and total protein concentration was measured using Bradford’s assay [16]. Each 10 μL of standard BSA (0.0625, 0.125, 0.25, 0.5, 1, 1.5 and 2 mg/mL) or sample solution was mixed with 1 mL of working Bradford reagent (5× dilution of the stock Bradford solution). The mixtures were mixed well and incubated in dim light at room temperature for 10 min and the optical density at 595 nm was determined [17].

### 2.4. Western Blot Analysis of the Recombinant ERFE–FLAG Expression 

The culture media (1 mL) was concentrated on a membrane filter unit (Amicon^®^ Ultra-15, molecular weight (MW) cutoff of 10 kD, Merck KGaA, Darmstadt, Germany) by centrifugation at 12,000 rpm for 10 min and was loaded into each well of the 10% SDS-PAGE gel along with MW marker and run at 300 V, 25 mA for 1.5 h. The proteins from the gel were then transferred to nitrocellulose membrane at 30 V, 400 mA for 1.5 h, and stained with Ponceau-S dye solution. Next, the membrane was blocked with 5% (*w*/*v*) skim milk which had been previously prepared in PBS containing 0.05% Tween-20 (PBST) for 1 h at room temperature. The rabbit anti-FLAG IgG antibody (1:1000) was applied to the membrane which was incubated overnight at 4 °C and washed three times with skim milk. The membrane was incubated with the goat anti-rabbit IgG antibody conjugated with HRP (1:10,000) at room temperature for 1 h, washed with skim milk three times, and developed for signal using Pierce™ ECL Western Blotting Substrate following the kit manufacturer’s instructions [16].

### 2.5. Purification of ERFE–FLAG Protein

The expressed ERFE–FLAG protein was purified by the immuno-precipitation technique using anti-FLAG IgG affinity resin. The eluted ERFE–FLAG protein was confirmed by western blot analysis compared to the eluted mock sample [16].

### 2.6. Treatment of HepG2 and Huh7 Cells with ERFE

In time course, the cells were cultured in rERFE–FLAG (50, 100 and 150 μg/mL)-rich DMEM or purified ERFE (0.75, 1.5 and 3 μg/mL)-supplemented DMEM for 1, 3, 6, 12, 24 and 48 h and hepcidin mRNA was measured using qRT-PCR [16]. For dose response, HepG2 and Huh7 cells were cultured in rERFE–FLAG (50, 100 and 200 μg/mL)-rich DMEM of HEK293 cells transfected with pcDNA3.1 + ERFE (50%, *v/v*) for 24 h and the cells were collected for measurement of hepcidin mRNA using the quantitative real-time polymerase chain reaction (qRT-PCR) method as described below. In another study, these two cells were cultured in purified ERFE (0.75, 1.5 and 3 μg/mL)-supplemented DMEM for 24 h and collected for measurement of hepcidin mRNA using qRT-PCR [16].

### 2.7. Quantitative Real-Time Polymerase Chain Reaction Asssay

Total RNA was extracted from the cells (HepG2 and Huh7) using TRIzol reagent according to the manufacturer’s protocol; concentration was determined with a Nano-Drop spectrophotometer at a wavelength ratio of 260:280., with all samples having a higher ratio than 1.8. Total RNA (1 μg) was then used for synthesizing the complementary DNA of hepcidin (*HAMP*) and ribosomal protein L19 (*RPL19*) genes by using a High Capacity cDNA Reverse Transcription Kit, according to manufacturer’s protocol. Finally, the qRT-PCR results were proceeded using a specific primer sequence for the *HAMP* gene with EXPRESS SYBR GreenER qPCR Supermix Universal Kit on the ABS 7500 fast real-time PCR machine, according to manufacturer’s instruction [16]. The *HAMP* expression was normalized with constitutive *RPL19* and expressed as fold changes using (−delta^2^ CT).

### 2.8. Statistical Analysis

The data are presented as mean ± standard deviation (SD) and were analyzed using the IBM^®^ SPSS^®^ Statistics V22.0 Program (IBM Company, Chicago, IL, USA). Statistical significance was determined using Student’s *t*-test; *p* < 0.05 was considered significant.

## 3. Results

### 3.1. Expression of ERFE–FLAG

The western blot results revealed that recombinant erythroferrone-FLAG (ERFE–FLAG) protein was successfully expressed in HEK293 cells and secreted into the medium, showing an intense protein band with a molecular weight (MW) of 37 kD (Figure 1). Though the estimated ERFE protein size is about 37 kD according to amino acid composition, the actual band size is larger than expected because of its three glycosylation sites. In this study, secreted ERFE–FLAG was selected to be used in our experiments because its glycosylation sites might be involved in the binding of its cognate receptor. The smaller unglycosylated faint band appears in the cell pellet lines because the ERFE protein might not have been glycosylated. In addition, the result confirms that the expressed ERFE–FLAG protein was successfully purified and concentrated from the culture medium (Figure 2).

### 3.2. Hepcidin mRNA Expression in Hepatoma Cells at Different Time Points

Interestingly, the hepcidin mRNA levels of HepG2 cells cultured in 50% (*v/v*) DMEM of HEK293 cells transfected with pcDNA3.1 + ERFE were significantly decreased at 3 and 6 h when compared to of HEK293 cells transfected with pcDNA3.1+, and restored to the original level at 12–48 h (Figure 3A). In comparison, hepcidin mRNA levels of HepG2 cells cultured in 3 μg/mL purified ERFE-supplemented DMEM were initially decreased at 1 h, then restored to the original levels at 3 and 6 h, and tended to increase at 12 and 24 h when compared to the control cell (Figure 3B).

Similarly, the hepcidin mRNA levels of Huh7 cells cultured in 50% (*v/v*) DMEM of HEK293 cells transfected with pcDNA3.1 + ERFE were significantly decreased at 3 and 6 h when compared to HEK293 cells transfected with pcDNA3.1+, and restored to the original level at 24–48 h (Figure 4A). Surprisingly, the hepcidin mRNA levels of Huh7 cells cultured in 3 μg/mL purified ERFE-supplemented DMEM were significantly decreased at 1 and 3 h when compared to the control cell, and rapidly increased to higher levels at 12 and 24 h (*p* < 0.05) (Figure 4B).

### 3.3. Hepcidin mRNA (hamp1) Expression in Hepatoma Cells at Different ERFE Concentrations 

The results have revealed that hepcidin mRNA levels were not changed in HepG2 cells after treatments with different concentrations of ERFE–FLAG existing in the DMEM of transfected HEK293 cells for 24 h (Figure 5A). Interestingly, hepcidin mRNA levels were significantly increased in the cells treated with purified ERFE (0.75–3 μg/mL)-supplemented DMEM, of which a dose of 0.75 μg/mL was the most effective (Figure 5B).

In comparison, hepcidin mRNA levels were not changed in Huh7 cells after treatment with different concentrations of ERFE–FLAG existing in the DMEM of transfected HEK293 cells for 24 h (Figure 6A). Nonetheless, hepcidin mRNA levels tended to increase in cells treated with purified ERFE (0.75 and 1.5 μg/mL)-supplemented DMEM (Figure 6B).

## 4. Discussion

In this study, pcDNA3.1 + ERFE-flag plasmid was determined to express erythroferrone protein because it has eight amino acids (DYKDDDDK) which are tagged to the C-terminal of ERFE, and which thus have very little chance of interfering with the folding of ERFE. In our experiment, recombinant ERFE-flag protein size was larger than expected because extracellular ERFE contains three distinct glycosylation sites. Seldin and co-workers also used pcDNA3.1 + ERFE-flag for mouse ERFE in their experiment and showed that ERFE enhances lipid uptake [18,19]. Although the human primary liver cells are the best for identification of ERFE receptors, ethical considerations present a great hindrance for experiments. Therefore, human cancerous liver cell lines still needed to be screened. In general, the secondary cancer cells differ from primary cells either slightly or significantly. Some cancer cell lines change phenotypically and lose their normal receptors on the membrane [19]. Furthermore, the metabolism of cancer cell lines is very different from that of normal primary cells and sometimes does not reflect normal human conditions. Therefore, some drug metabolism and pharmacological effects are needed to measure nearly-normal liver cell lines such as THLE-2, THLE-3, etc., which are different from cancerous cell lines [20].

The two functions of ERFE are to suppress hepcidin expression and to increase lipid uptake [1,5,19]. Based on the suppressive action of ERFE on the hepcidin gene and the protein nature of ERFE hormone, the ERFE receptor should be mostly on liver cell membranes, where it is used as a separate signaling cascade to act on hepcidin gene expression [1,11]. However, the limitations on the use of primary liver cells directs this experiment to screen the liver cell lines that have ERFE receptors. Although there are a vast number of liver cell lines to be screened, HepG2 and Huh7 cells were selected in this experiment because these cell lines are very common and were easily available to us.

The actual concentration of ERFE protein needed to suppress hepcidin transcription and the time duration for ERFE treatment is still unknown. The recombinant ERFE–FLAG concentration needed is more than 5 μg/mL in order to be significantly involved in increased lipid uptake processes in mouse 3T3-L1 adipocytes and mouse H4IIE hepatocytes after 12 h [18]. However, the blood hormone concentration in humans is very low, usually on the nanogram to picogram scale. The normal blood ERFE concentration is 42.22 ± 16.55 pg/mL in healthy children and 197.00 ± 185.51 pg/mL in iron-deficient anemic children [21]. Leon Kautz and his colleagues used 50% of culture media from transfected 293T cells with pcDNA3.1 + mouse ERFE–FLAG to treat mouse primary liver cells and measured hepcidin mRNA levels after 15 h of treatment; hepcidin mRNA levels were significantly suppressed [1]. Therefore, we decided to use 50% of cell culture media of transfected 293 cells with pcDNA3.1 + ERFE-flag. 

Previous studies have revealed the effect of ERFE on hepcidin as being hepcidin-suppressive. For instance, ERFE was found to suppress hepcidin expression and secretion, consequently resulting in increased iron absorption during instances of stress erythropoiesis (e.g., acute blood loss) [1]. Mutations of the erythroferrone gene (ERFE A260S) have been found in congenital dyserythropoietic anemia type II patients (12.5%), wherein ineffective erythropoiesis can lead to increased ERFE production, decreased hepcidin secretion, and ultimately iron loading [22]. A recent study in mice has shown that a single dose of EPO increased the expression and production of ERFE in the erythroblast of Fam132b, transferrin receptor-1 (TfR1), and -2 (TfR2) in the spleen while decreasing hepatic hepcidin mRNA expression [23]. Nevertheless, serum levels of ERFE and EPO, along with erythropoietic activity and hepcidin, were higher in active rheumatoid arthritis patients who had anemia than in control patients. This would seem to indicate that ERFE and hepcidin may be independent predictors [24]. Moreover, there was a positive correlation between serum hepcidin concentrations and divalent metal ion transporter-1 (DMT1) and TfR1 in the liver, as well as in serum ERFE concentrations and TfR2 in the liver [25]. In our experiments, hepcidin was significantly suppressed by the recombinant ERFE-flag protein at specific time points (3 and 6 h) in Huh7 cells treated with both 50% (*v/v*) ERFE-containing DMEM and 3 μg/mL, but not in HepG2 cells. The different response of Huh7 and HepG2 to ERFE protein might possibly be due to the existence of different phenotypical changes such as the loss of normal receptors on their membranes or the inconsistent production of the ERFE binding co-protein. Consequently, Huh7 might be the most suitable cell line for use in ERFE treatment experiments. The ERFE effect on hepcidin gene suppression optimally occurs within 3–6 h after treatment, while loss of the suppressive effect occurs within 24–48 h. In the future, when using recombinant ERFE–FLAG protein and Huh7 cells, the ERFE receptor or its cognate protein could be pulled down by protein interaction through co-immunoprecipitation and then identified by mass spectrophotometry analysis.

## 5. Conclusions

Human ERFE was successfully expressed as a FLAG-tagged protein at the C-terminal region in HEK293 cells. Two common liver cell lines (HepG2 and Huh7) were screened to investigate the suppressive action of recombinant ERFE-flag protein. Only Huh7 showed hepcidin suppression at time point 6 h. Therefore, Huh7 and recombinant ERFE–FLAG can be used in future experiments to identify the ERFE receptor via protein interaction such as co-immunoprecipitation and mass spectrometry analysis.

## Figures and Tables

**Figure 1 materials-14-06480-f001:**
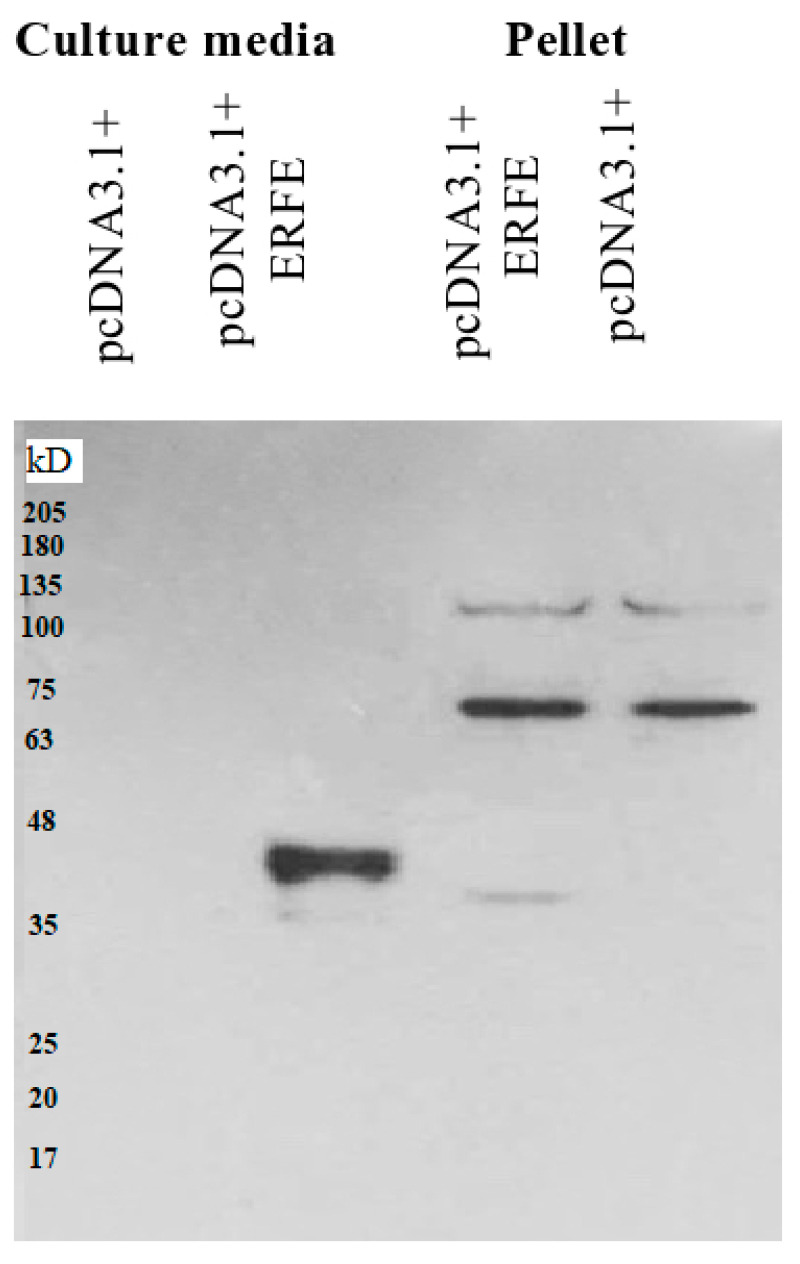
Western blot analysis of ERFE–FLAG protein expression in culture medium and pellet of HEK293 cells transfected with pcDNA3.1 + ERFE–FLAG plasmid. HEK293T cells were transfected with pcDNA3.1 + ERFE–FLAG plasmid or standard pcDNA3.1 + (mock) and cultured in 10% FBS-supplemented DMEM for 24 h. The expressed ERFE–FLAG protein was analyzed in the medium and the cells using Western blot technique. Abbreviations: DMEM = Dulbecco’s Modified Eagle Medium, ERFE = erythroferrone, FBS = fetal bovine serum, HEK293 = human embryonic kidney 293.

**Figure 2 materials-14-06480-f002:**
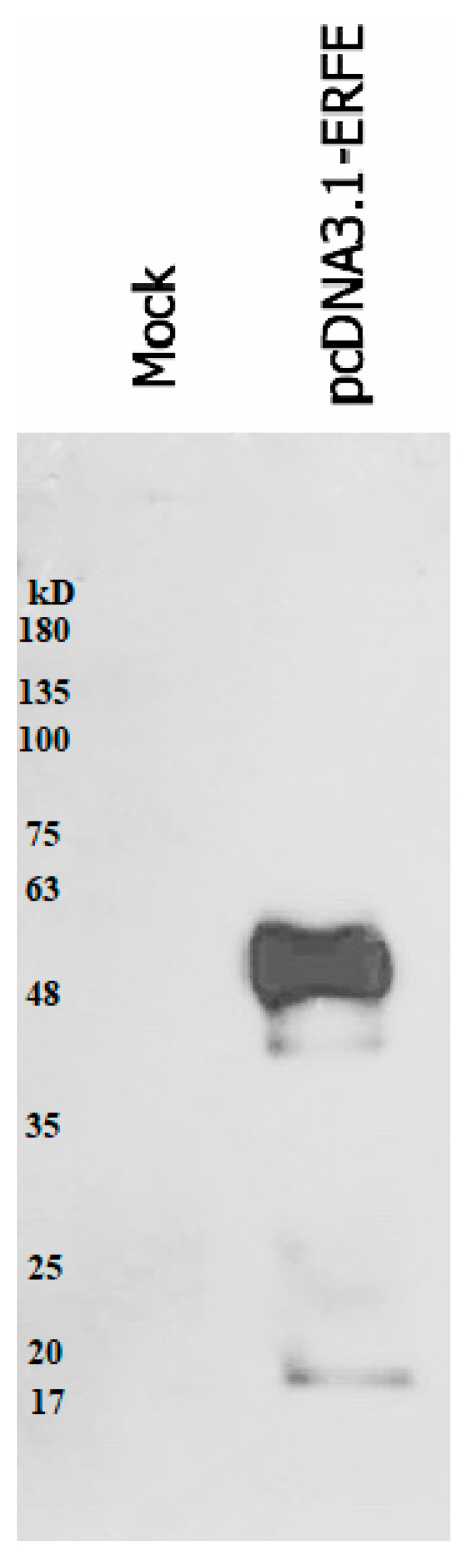
Western blot analysis of expressed ERFE–FLAG protein purified from the medium of HEK293 cells transfected with pcDNA3.1 + ERFE–FLAG plasmid. HEK293T cells were transfected with pcDNA3.1 + ERFE–FLAG plasmid or standard pcDNA3.1 + (mock) and cultured in 10% FBS-supplemented DMEM for 24 h. The expressed ERFE–FLAG protein was purified using immuno-precipitation technique and analyzed using Western blot technique. Abbreviations: DMEM = Dulbecco’s Modified Eagle Medium, ERFE = erythroferrone, FBS = fetal bovine serum, HEK293 = human embryonic kidney 293.

**Figure 3 materials-14-06480-f003:**
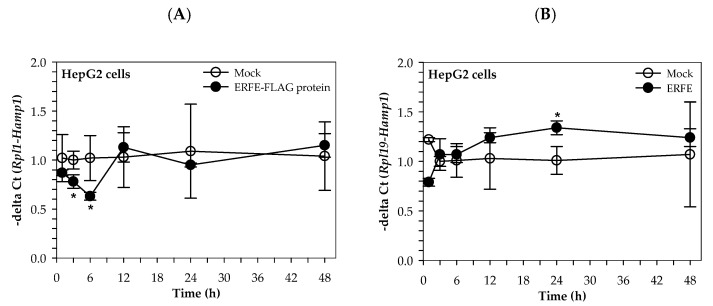
Relative hepcidin mRNA levels in HepG2 cells at various time points after ERFE treatment. HepG2 cells were cultured in 50% (*v/v*) DMEM obtained from HEK293 cells transfected with pcDNA3.1 + ERFE (ERFE–FLAG protein) or pcDNA3.1+ (MOCK) (**A**) or in 3 μg/mL purified ERFE-supplemented DMEM (**B**) for 1–48 h and determined mRNA levels using qRT-PCR. Data which were obtained from four repetitions and normalized with constitutive *RPL19* are expressed as mean ± SD of fold-change hepcidin mRNA expression using −delta Ct (*Rpl19-Hamp*). * *p* < 0.05 when compared with MOCK. Abbreviations: Ct = control, DMEM = Dulbecco’s Modified Eagle Medium, ERFE = erythroferrone, HEK293 = human embryonic kidney 293 cells, qRT-PCR = quantitative real-time polymerase chain reaction.

**Figure 4 materials-14-06480-f004:**
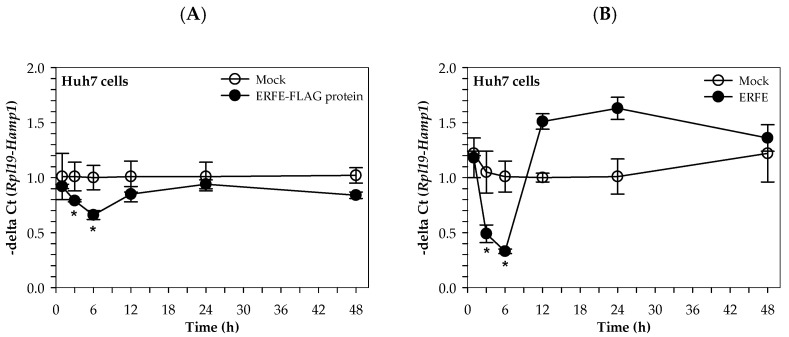
Relative hepcidin mRNA levels in Huh7 cells at various time points after ERFE treatment. Huh7 cells were cultured in 50% (*v/v*) DMEM obtained from HEK293 cells transfected with pcDNA3.1 + ERFE (ERFE–FLAG protein) or pcDNA3.1+ (MOCK) (**A**) or in 3 μg/mL purified ERFE-supplemented DMEM (**B**) for 1–48 h and determined mRNA levels using qRT-PCR. Data which were obtained from four repetitions and normalized with constitutive *RPL19* are expressed as mean ± SD of fold-change hepcidin mRNA expression using −delta Ct (*Rpl19-Hamp*). * *p* < 0.05 when compared with MOCK. Abbreviations: Ct = control, DMEM = Dulbecco’s Modified Eagle Medium, ERFE = erythroferrone, HEK293 = human embryonic kidney 293 cells, qRT-PCR = quantitative real-time polymerase chain reaction.

**Figure 5 materials-14-06480-f005:**
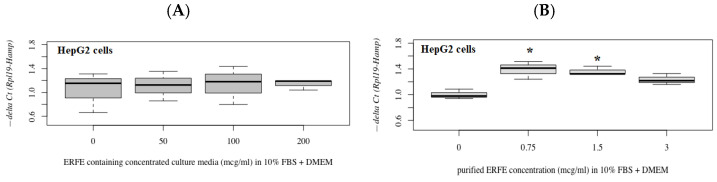
Relative hepcidin mRNA levels in HepG2 cells after 24 h of ERFE treatment. HepG2 cells were cultured in ERFE–FLAG-rich DMEM of the transfected HEK293 cells (**A**) or purified ERFE-supplemented DMEM (**B**) and mRNA levels determined using qRT-PCR. Data on fold changes in hepcidin mRNA expression using −delta Ct (*Rpl19-Hamp*) were obtained from four repetitions and normalized with constitutive *RPL19* are expressed as fold changes and as mean ± SD. * *p* < 0.05 when compared to no treatment. Abbreviations: Ct = control, DMEM = Dulbecco’s Modified Eagle Medium, ERFE = erythroferrone, FBS = fetal bovine serum, HEK293 = human embryonic kidney 293 cells, qRT-PCR = quantitative real-time polymerase chain reaction, RPL19 = ribosomal protein L19.

**Figure 6 materials-14-06480-f006:**
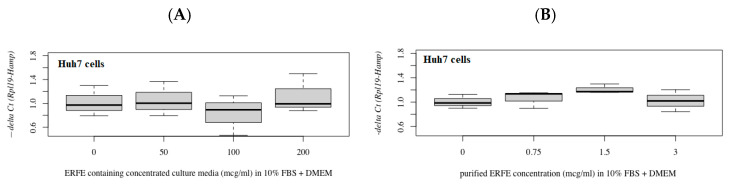
Relative hepcidin mRNA levels in Huh7 cells after 24 h of ERFE treatment. Huh7 cells were cultured in the ERFE–FLAG-rich DMEM of transfected HEK293 cells (**A**) or purified ERFE-supplemented DMEM (**B**) and mRNA levels determined using qRT-PCR. Data were obtained from four repetitions and normalized with constitutive *RPL19*, and are expressed as mean ± SD of fold-change hepcidin mRNA expression using −delta Ct (*Rpl19-Hamp*). *p* < 0.05 when compared to no treatment. Abbreviations: Ct = control, DMEM = Dulbecco’s Modified Eagle Medium, ERFE = erythroferrone, FBS = fetal bovine serum, HEK293 = human embryonic kidney 293 cells, qRT-PCR = quantitative real-time polymerase chain reaction.

## Data Availability

Data is contained within the article.

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
