# Peer review of "Effect of Recombinant Human Erythroferrone Protein on Hepcidin Gene (Hamp1) Expression in HepG2 and HuH7 Cells"

_materials, 2021, doi:10.3390/ma14216480_

Round 1

Reviewer 1 Report

  1. Hamp1 was descripted in title and abstract, however, hepcidin was wrote in introduction, result, discussion and conclusion sections. Though Hamp1 is hepcidin, authors have to uniform the noun.
  2. Many studies have demonstrated the erythroferrone may influence hepcidin expression. Authors have to introduce more relationships between erythroferrone and hepcidin referred to previous studies.
  3. Authors have to discuss the differences between previous studies and this study about how erythroferrone regulated hepcidin.
  4. Why the ERFE-FLAG have different molecular weight in figure 1? In addition, authors have to mention why the ERFE-FLAG obtained from medium was selected to treat HepG2 and Huhc 7cells.
  5. The figure 4 legend “Relative hepcidin mRNA levels in HepG2 cells at various” was a mistake. Please replace hepG2 with Huhc.

     6. Different effects on ERFE-FLAG-treated HepG2 and Huch 7 cells.        Authors have to discuss the possible reasons.

Reviewer 2 Report

The authors presented the influence of recombinant human erythroferrone on expression of hepcidin in hepatocellular carcinoma cell lines HepG2 and Huh7. They found that only in Huh7 cell line ERFE suppresses hepcidin expression what they ascribed to supposed ERFE receptors. In introduction should be I think presented previous knowledge justifying examinations undertook by authors what was not showed clearly. ERFE receptors are only supposed but not demonstrated by authors. Arezes et al. showed that secreted ERFE binds BMP and its receptor and this way inhibits hepcidin expression, may be it was this way in authors experiment. 

Summing up I think in introduction and in discussion should be shown more clearly why authors did the experiments and what comes out of it for the present and future.
